# Evaluation of Irrigation Modes for Greenhouse Drip Irrigation Tomatoes Based on AquaCrop and DSSAT Models

**DOI:** 10.3390/plants12223863

**Published:** 2023-11-15

**Authors:** Jiankun Ge, Zihui Yu, Xuewen Gong, Yinglu Ping, Jinyao Luo, Yanbin Li

**Affiliations:** 1College of Water Conservancy, North China University of Water Resources and Electric Power, Zhengzhou 450045, China; gejiankun@ncwu.edu.cn (J.G.); yuzihui1006@163.com (Z.Y.); liyb101@sina.com (Y.L.); 2Ningbo Water Conservancy and Hydropower Planning Design Institute Co., Ltd., Ningbo 315192, China; pyl1326933100@163.com; 3College of Water Resources and Hydropower, Wuhan University, Wuhan 430072, China; luojy@whu.edu.cn

**Keywords:** solar greenhouse, tomato, AquaCrop model, DSSAT model, drip irrigation, plastic mulching

## Abstract

The improvement of the simulation accuracy of crop models in different greenhouse environments would be better applied to the automation management of greenhouse cultivation. Tomatoes under drip irrigation in a greenhouse were taken as the research object, and the cumulative evaporation capacity (*Ep*) of the 20 cm standard evaporation dish was taken as the basis for irrigation. Three treatments were set up in the experiment: high water treatment without mulch (NM-0.9 *Ep*), high water treatment with mulch (M-0.9 *Ep*), and low water treatment with mulch (M-0.5 *Ep*). AquaCrop and DSSAT models were used to simulate the canopy coverage, soil water content, biomass, and yield of the tomatoes. Data from 2020 were used to correct the model, and simulation results from 2021 were analyzed in this paper. The results showed that: (1) Of the two crop models, the simulation accuracy of the greenhouse tomato canopy coverage *k_CC_* was higher, and the root mean square errors were less than 6.8% (AquaCrop model) and 8.5% (DSSAT model); (2) The AquaCrop model could accurately simulate soil water change under high water treatments, while the DSSAT model was more suitable for the conditions without mulch; (3) The relative error RE of simulated and observed values for biomass B, yield Y, and water use efficiency WUE in the AquaCrop model were less than 2.0%, 2.3%, and 9.0%, respectively, while those of the DSSAT model were less than 4.7%, 7.6%, and 10.4%, respectively; (4) Considering the simulation results of each index comprehensively, the AquaCrop model was superior to the DSSAT model; subsequently, the former was used to predict 16 different water and film coating treatments (S1–S16). It was found that the greenhouse tomato yield and WUE were the highest under S7 (0.8 *Ep*), at 8.201 t/ha and 2.79 kg/m^3^, respectively.

## 1. Introduction

Greenhouse cultivation is an essential planting mode to ensure the safety and stability of China’s “food basket project” [1]. At this stage, greenhouse development is getting bigger and bigger. However, in many areas of greenhouse cultivation, there are still problems, such as waste of water resources, decrease in yield and quality due to unscientific management of irrigation, and so on [2].

Crop growth models are powerful tools for developing irrigation and fertilization schedules and predicting yields [3]. In order to realize water-saving and efficient agricultural production, many scholars have conducted a large number of studies on the growth and development [4], water consumption patterns [5], yield [6], and water use efficiency of crops [7] under different cultivation and management schedules by using crop models, thus reducing unnecessary human and material inputs in field trials. For example, Battisti et al. [8] simulated the growth and development of soybean using five crop growth models and conducted a comparative analysis of the simulation results of each model, concluding that model integration was better than using only a single model. Martre [9] analyzed 27 crop growth models and discussed the evaluation of different models for simulation of canopy cover and soil water content in wheat, and the results showed that the AquaCrop and DSSAT models were more capable of accurately simulating canopy cover and soil water content in wheat than other models. Ventrella D et al. [10] used the DSSAT model to assess the production potential of wheat and tomatoes in Italian regions under future climatic conditions and found that when the sowing of wheat was delayed, the yield was not negatively affected by climate change, but early transplanting of tomatoes minimized the impact of climate change on crop productivity. Some scholars in China have evaluated the applicability of the AquaCrop model for crops such as wheat, maize, and grapes, involving Guanzhong Plain [11], Jinzhong Basin [12], Songnen Plain [13], and Loess Plateau, etc. [14]. The simulation accuracy was high, and it was agreed that AquaCrop [15] balanced the simulation accuracy of the crop model, the simplicity of the input parameters, and the stability of simulation results compared with other crop models. In addition, Yao Ning et al. [16] used the DSSAT model to validate the simulation of drought-affected winter wheat during the tasseling and grouting periods, and found that the simulation results of crop growth and development and soil moisture change processes were better, but the simulation accuracy would be gradually reduced with the increase of the degree of drought. Zou Long et al. [17] used the DSSAT model to simulate spring maize under different water and fertilizer conditions, and the results showed that high fertilizer and water conditions were more able to promote the formation of yield. Haidong Wang et al. [18] used AquaCrop and DSSAT-SUBSTOR-Potato models to simulate the canopy cover, yield, and water production and utilization efficiency of potato in the sandy area of Northwest China, and the results showed that the simulation of the DSSAT-SUBSTOR-Potato model was better under the high water conditions, but, overall, its simulation accuracy was lower than that of the AquaCrop model. It can be seen that the AquaCrop model and the DSSAT model [19] have been evaluated for performance on different regions and crop types with high simulation accuracy. They are the most applied crop growth models at present. However, under different management systems (mulching and irrigation) of greenhouse crops, research on the applicability of the AquaCrop model parameters, simulation accuracy, and other related issues has not been systematic enough, and there are few comparative studies on the applicability of the two models. In summary, this study conducted a two-year experiment in a solar greenhouse to study greenhouse drip-irrigated tomatoes in order to achieve the following objectives: (1) Optimize the parameters of the AquaCrop and DSSAT models to improve the accuracy of modeling greenhouse tomatoes in North China; (2) Simulate canopy coverage, soil moisture, yield, biomass, and water use efficiency of tomatoes under different water and mulching conditions using the optimized model parameters; (3) Evaluate the applicability of the AquaCrop and DSSAT models for greenhouse tomatoes in North China; (4) Explore the optimal combination of “irrigation level + mulching” for water management, providing reference for further optimization of the management system for greenhouse tomato cultivation.

## 2. Materials and Methods

### 2.1. Overview of the Experimental Area

The experiments were conducted in the solar greenhouse of the Xinxiang Experimental Base of the Chinese Academy of Agricultural Sciences in 2020 and 2021 (35°9′ N, 113°5′ E, elevation 78.7 m). The solar greenhouse is oriented in an east-west direction and faces south, with a total area of 510 m^2^ (60 m in length and 8.5 m in width). The top of the greenhouse was covered with non-drip polyethylene film (0.2 mm thick) and insulation cotton (5 cm thick). The walls of the greenhouse are 60 cm thick and contain insulation materials to enhance insulation. The soil in the experimental area, from a 0 to 100 cm depth, is loam, with an average bulk density of 1.48 g/m^3^. The physicochemical properties and characteristic parameters of the soil are shown in Table 1.

The tomato variety used in the experiment was “Jinpeng M6”. The transplanting dates were 8 March 2020 and 8 March 2021. The planting density was 5.7 plants per square meter. The basal fertilizer consisted of 112 kg/ha urea (containing 46% N), 150 kg/ha potassium sulfate (containing 50% K_2_O), and 120 kg/ha calcium superphosphate (containing 14% P_2_O_5_). After starting water treatments, integrated fertilizer application was performed using a water-fertilizer integrated system during the 2nd, 4th, 6th, 8th, and 10th irrigation, with 18.8 kg/ha urea and 25 kg/ha potassium sulfate applied at each time. After tomatoes set fruit, 5 layers of fruit were reserved, with 4 fruits per layer. All agronomic measures, such as topping and spraying, were the same for all plots.

### 2.2. Experimental Design

The cumulative evaporation from a 20 cm standard evaporation dish (*Ep*) was used as the basis for irrigation. Three combinations of two water levels and two mulching treatments were set: No mulch high water (NM-0.9 *Ep*), Mulch high water (M-0.9 *Ep*), and Mulch low water (M-0.5 *Ep*). The experiment was designed using a randomized complete block design with three replicates for each treatment. The size of each experimental plot was 8.0 m in length and 2.2 m in width. Wide-narrow row planting with an alternating pattern was adopted, with a wide row spacing of 65 cm, narrow row spacing of 45 cm, and plant spacing of 30 cm. Plastic film was buried between the plots to a depth of 60 cm. The irrigation method used was drip irrigation, with drip emitters having a flow rate of 1.1 L/h and spaced at 30 cm intervals, corresponding to the plants. The evaporation dish was placed 30 cm above the canopy of the plants and was adjusted according to the growth of the tomatoes. The evaporation was measured daily using a graduated cylinder with a precision of 0.1 mm at 7:30 to 8:00. Irrigation was conducted when the cumulative evaporation (*Ep*) reached 20 ± 2 mm [20]. The irrigation quota (*I_r_*) was calculated using the following formula:(1)$$I_{r}=Ep\times K_{c}$$
where $I_{r}$ represents the irrigation quota (mm); $K_{c}$ represents the water surface evaporation coefficient; and Ep represents the cumulative evaporation (mm).

After transplanting, tomatoes were supplemented with 20 mm of water for seedling establishment. Once the tomatoes entered the rapid growth phase, water management treatments began.

### 2.3. Measurement Items and Methods

#### 2.3.1. Meteorological Data

The meteorological data for the tomato growing season at the Xinxiang Experimental Base in 2020 and 2021 were obtained from a fully automated weather station located inside greenhouse. The continuous monitoring period was set from 4 March to 10 July in 2020 and from 5 March to 8 July in 2021. The system included a set of net radiation sensors (Rn, NRLITE2, Kipp & Zomen, Delft, The Netherlands), total radiation sensors (Rs, LI200X, Campbell Scientific, Inc., Logan, UT, USA), and temperature and relative humidity sensors (Ta, RH., CS215, Campbell Scientific, Inc., USA). The weather station was installed at a height of 2 m above the ground level, with 5 sets of sensors. The wind speed was monitored by a WindSonic anemometer (u_2_, WindSonic, Gill, UK), located at 2 m above the ground surface, with an accuracy of ±0.02 m/s. Soil heat flux (G) was measured by inserting a heat flux plate (HFP01, Hukseflux, Delft, The Netherlands) between the soil surface and two tomato plants, at a depth of 5 cm. All data were collected every 5 s and averaged over 30 min, and were recorded in the CR1000 data acquisition system (Campbell Scientific Inc., USA).

#### 2.3.2. Soil Water Content

Soil water content was measured at the intermediate position between two representative plants [21]. The soil water profile was determined using a TRIME-IPH time-domain reflectometer (IMKO, Ettlingen, Germany) at depths of 20, 40, and 60 cm. Measurements were conducted every 7 days throughout the whole growth period, with three replicates for each treatment. The moisture content values were averaged. The TRIME-IPH instrument was periodically calibrated using the soil-drying method.

#### 2.3.3. Leaf Area Index and Canopy Cover

Ten healthy plant specimens with uniform growth and no pests or diseases were randomly selected and marked in each plot. The leaf area (leaf length L and maximum width W_m_) was measured using a ruler at intervals of 7–10 days [22] and was calculated using the reduction factor method, with a conversion factor of 0.685.

Crop canopy coverage (*k_CC_*) was calculated using the following formula:(2)$$k_{CC}=1-e^{-C\cdot LAI}$$
where $k_{CC}$ is the canopy coverage; C is the extinction coefficient, with value s 0.8; and LAI is the leaf area index.

#### 2.3.4. Yield, Biomass, and Water Use Efficiency

At the end of the experiment, 5 representative plants from each plot were selected. The above-ground parts of the tomato plants from each treatment were placed in an oven at 105 °C for 30 min to kill the plant tissue, and then they were dried at 75 °C until a constant weight was achieved. The dried biomass (B) was measured. During the final yield measurement, 3 representative fruits from each plot were selected. The fresh weight of the fruits was measured, and then the fruits were sliced using a quartering method. The sliced fruits were placed in an oven at 105 °C for 30 min to kill the tissue, and then they were dried at 75 °C until a constant weight was achieved. The dry weight (Y) of the fruits was measured, with 3 replicates for each treatment.

The formula for calculating water use efficiency is as follows: (3)$$WUE=Y/ET_{c}\times 100$$
where WUE represents water use efficiency (kg/m^3^); Y represents yield (t/ha); and $ET_{C}$ represents actual crop evapotranspiration (mm).

## 3. Model and Parameters Calibration

### 3.1. AquaCrop Model

#### 3.1.1. Model Introduction

The AquaCrop model divides evapotranspiration (ET) into two components: E_s_ and *T_r_*. It incorporates the crop harvest index (HI) to adjust the proportion of biomass (B) produced. The core equation of the AquaCrop is as follows [23]:(4)Y=B⋅HI
(5)$$B=WP\cdot \sum T_{r}$$
where Y represents total yield (Kg/m^2^); B represents total biomass (Kg/m^2^); HI represents crop harvest index (%); $T_{r}$ represents crop transpiration (mm); and WP represents water productivity of biomass (kg/(m^2^·mm)).

#### 3.1.2. Parameter Configuration

Meteorological Data: The meteorological data used in this model for the years 2020–2021 were obtained from a fully automated weather station located inside the greenhouse. This data included rainfall, daily maximum temperature, and daily minimum temperature. The experiments were conducted inside the greenhouse with no rainfall.

The reference crop evapotranspiration (ET_0_) was based on the Penman-Montieth model with a revised aerodynamic resistance parameter (r_a_) of 295 s/m, as described by Fernández [24]. The calculation formula for ET_0_ is as follows:(6)$$ET_{0}=\frac{0.408\Delta (R_{n}-G)+\gamma \frac{628(e_{s}-e_{a})}{T+273}}{\Delta +1.24\gamma }$$
where $ET_{0}$ represents reference crop evapotranspiration (mm/d); $R_{n}$ represents net radiation at the crop surface (MJ/m^2^/d); G represents soil heat flux (MJ/m^2^/d); T represents the daily average temperature at a height of 2 m above the ground (°C); $e_{s}$ represents the saturated vapor pressure (kPa); $e_{a}$ represents the actual vapor pressure (kPa); $e_{s}-e_{a}$ represents the vapor pressure deficit (kPa); Δ represents the slope of the saturation vapor pressure-temperature curve; and γ represents the psychrometric constant (kPa/°C).

The air temperature, solar radiation, and ET_0_ for the years 2020 and 2021 are shown in Figure 1.

During the entire growth period of the tomatoes in 2020 and 2021, the indoor temperature (T_a_) reached a maximum of 40.17 °C and 42.89 °C, with minimum values of 9.14 °C and 9.56 °C, and average values of 23.44 °C and 24.64 °C, respectively. The maximum solar radiation (R_s_) for the two years were 14.64 MJ·m^−2^·d^−1^ and 16.14 MJ·m^−2^·d^−1^, with minimum values of 0.50 MJ·m^−2^·d^−1^ and 1.37 MJ·m^−2^·d^−1^, respectively. The average solar radiation values were 8.89 MJ·m^−2^·d^−1^ and 8.26 MJ·m^−2^·d^−1^, respectively. The maximum ET0 values for the two years were 5.22 mm and 5.46 mm, with minimum values of 0.36 mm and 0.30 mm, and average values of 2.67 mm and 2.33 mm, respectively.

Management data: The irrigation method (drip irrigation), plastic film covering (with film, without film), and other management parameters were set based on the actual conditions of the tomato experiment.

Crop data: The crop parameters inputted into the model include the growth status of tomatoes, canopy growth status, maximum effective root depth, types of stress the crop experiences, and factors affecting the stress. A crop parameter database file was generated by inputting the actual growth and development of the tomatoes and recommended values from the model.

Soil data: The soil texture in the experimental area was loam from 0 to 100 cm. The soil depth was 100 cm, divided into 5 layers, with each layer having a thickness of 20 cm. Soil physical and chemical parameters were shown in Table 1. Soil depth and initial soil water content were input based on the tomato experimental data to generate the initial conditions for running the model.

#### 3.1.3. Parameter Calibration

In order to improve the simulation accuracy of the model, after constructing the model database, parameter calibration was necessary [25]. In this study, field experimental data from three treatments in 2020 were used for parameter calibration. For the parameters that needed to be adjusted, the recommended values for tomatoes provided by the AquaCrop model were referenced. The parameter adjustment range was controlled within 5% using a trial-and-error method. After establishing and generating the simulation control file, simulations were conducted to analyze the differences between simulated values and measured values. Then, model parameter values were adjusted to achieve the best simulation performance. The calibrated parameters were used to simulate the three treatments in 2021 and the results of the calibrated parameters were validated. The parameters of the AquCrop model after tau correction were shown in Table 2.

### 3.2. DSSAT Model

#### 3.2.1. Model Introduction

The Decision Support System for Agrotechnology Transfer (DSSAT) is one of the most widely used crop models worldwide [26]. It has the capability to simulate the growth, development, yield, irrigation scheduling, and fertilizer management of various crops [27]. DSSAT plays a crucial role in guiding field management, aiding in the determination of crop management decisions, and facilitating agricultural technology transfer [28].

#### 3.2.2. Parameter Configuration

Meteorological data: The parameters required to input into the DSSAT model’s meteorological database included solar radiation, daily maximum temperature, and daily minimum temperature. The meteorological data used for the model in 2020–2021 were obtained from an automated weather station inside the greenhouse. Since the experiments were conducted inside the greenhouse for two years, the rainfall amount was considered as 0.

Management data: Field management parameters included transplanting date, planting density, irrigation amount and timing, and fertilizer application amount and timing for tomatoes.

Crop data: In this study, reference was made to the work of Zhao Zilong [29] to select 10 variety parameters describing tomatoes. Afterwards, measured data such as yield, leaf area index, biomass, etc., were used, and the parameters were calibrated and adjusted using the embedded GLUE module in the model.

Soil data: Soil parameters consisted of basic physicochemical properties of the soil and data on soil profiles, including soil type, bulk density, soil particle composition, soil hydraulic parameters, etc. The soil profile parameters required by the model were determined based on the measured experimental data. The physical parameters of the soil are shown in Table 1.

Output parameters: The set output results in this study included leaf area index, soil moisture content, yield, and biomass.

#### 3.2.3. Parameter Calibration

Based on the recommended range or values for tomatoes provided by the DSSAT model, a simulation control file was established and generated for the 2020 experiment. Subsequently, using the measured yield data, the parameters were adjusted and calibrated using the GLUE module integrated in the model until the simulated values aligned well with the measured values. The calibrated parameter values were then used to simulate the tomato growth process for the same treatment in 2021. The simulated values for 2021 were compared and analyzed against the relevant measured data from the experiment. This process validated the results of the calibrated parameters. The calibrated parameters are displayed in Table 3.

### 3.3. Evaluation Indicators

Data analysis and chart plotting were performed using SPSS 26.0, Microsoft Excel 2010, and Origin 2022. Multiple comparisons were conducted using the Duncan analysis method of single-factor ANOVA. The accuracy of the model was evaluated using root mean square error (RMSE), normalized root mean square error (NRMSE), model efficiency index (EF), index of agreement (d_IA_), and relative error (RE). A smaller RMSE indicates better accuracy, while a NRMSE, RE, d_IA_, and EF closer to 0 and 1, respectively, represent higher model simulation accuracy.
(7)$$RMSE=\sqrt{\frac{\sum_{i=1}^{N}{\left(O_{i}-S_{i}\right)}^{2}}{N}}$$
(8)$$NRMSE=\frac{1}{\bar{O}}\sqrt{\frac{\sum_{0}^{N}(O_{i}-S_{i}{)}^{2}}{N}}\times 100\%$$
(9)$$EF=\frac{\sum_{1}^{N}{\left(O_{i}-\bar{O}\right)}^{2}-\sum_{1}^{N}{\left(S_{i}-O_{i}\right)}^{2}}{\sum_{1}^{N}{\left(O_{i}-\bar{O}\right)}^{2}}$$
(10)$$d_{IA}=1-\frac{\sum_{i=1}^{N}{\left(O_{i}-S_{i}\right)}^{2}}{\sum_{i=1}^{N}{\left(\left|S_{i}-\bar{O_{i}}\right|+\left|O_{i}-\bar{O_{i}}\right|\right)}^{2}}$$
(11)$$RE=\left|\frac{O_{i}-S_{i}}{O_{i}}\right|\times 100\%$$
where $O_{i}$ is the observed value; $S_{i}$ is the simulated value; $\bar{O}$ is the mean of the observed values; $\bar{S}$ is the mean of the simulated values; and N is the sample size.

## 4. Results

### 4.1. Canopy Cover

The simulated canopy cover of tomatoes under three cultivation modes in 2021 was obtained using the calibrated AquaCrop model and DSSAT model, as shown in Figure 2 and Figure 3. It can be observed that both models performed well in simulating canopy cover. As the growing stage processed, the simulated values of canopy coverage from both models had small deviations from the observed values in the early and late stages of growth, within 4% deviation. However, there was a slight deviation of approximately 10.2% observed during the rapid growth period (11–40 days after transplanting).

During the entire growth period, the AquaCrop model showed RMSE (Root Mean Square Error) values of 6.4%, 6.8%, and 5.0% for canopy cover in the NM-0.9 *Ep*, M-0.9 *Ep*, and M-0.5 *Ep* treatments, respectively. The consistency index (d) was 0.98 for all treatments. On the other hand, the DSSAT model exhibited RMSE values of 6.6%, 5.7%, and 8.5% for canopy cover in the NM-0.9 *Ep*, M-0.9 *Ep*, and M-0.5 *Ep* treatments, respectively, with *d_IA_* (index of agreement) values of 0.98, 0.98, and 0.99, respectively. These results indicate that the AquaCrop model performed better and had higher applicability in the NM-0.9 *Ep* and M-0.5 *Ep* environments, while the DSSAT model was more suitable for the M-0.9 *Ep* environment.

The evaluation indicators of simulated and observed values for canopy cover by the two models are shown in the Table 4.

### 4.2. Soil Water Content

Figure 4 shows the simulated results of soil moisture content at depths of 20 cm, 40 cm, and 60 cm under the three treatments, using the AquaCrop and DSSAT models.

The evaluation indicators of simulated and measured values of soil water content (SWC) by the two models in the three soil layers are shown in Table 5.

In Figure 4, it can be observed that the simulated values (Asim, Dsim) and observed values (obs) of soil water content at depths of 20 cm, 40 cm, and 60 cm exhibited significant fluctuations with different irrigation times during the entire growth period of the tomatoes. During the rapid growth and mid-growth stages, the plants required a large amount of water, resulting in an overall decreasing trend in soil water content. As the plants entered the later stages of growth, their physiological processes declined, and the leaves gradually yellowed, leading to a reduced water demand and a stabilized soil-water content.

Based Table 5, it can be observed that the AquaCrop model generally achieved good simulation results for soil water content (SWC), with lower RMSE and NRMSE values observed in the NM-0.9 *Ep* and M-0.5 *Ep* treatments. The DSSAT model exhibited lower RMSE and NRMSE values in the NM-0.9 *Ep* treatment compared to the other treatments. In the NM-0.9 *Ep* treatment, the DSSAT model showed slightly lower RMSE and NRMSE values for SWC at depths of 20 cm, 40 cm, and 60 cm compared to the AquaCrop model, with *d_IA_* values and EF closer to 1. On the other hand, the AquaCrop model had lower RMSE and NRMSE values compared to the DSSAT model in the M-0.9 Ep and M-0.5 *Ep* treatments, with *d_IA_* values and EF closer to 1. Therefore, it can be concluded that the AquaCrop model had higher simulation accuracy under the High water treatments (NM-0.9 *Ep* and M-0.9 *Ep*), followed by the Low water treatment (M-0.5 *Ep*). The DSSAT model demonstrated higher accuracy in the No mulch treatment (NM-0.9 *Ep*) and lower accuracy in the Mulch with low water treatment (M-0.5 *Ep*).

### 4.3. Biomass, Yield, and WUE

The comparative results between the simulated biomass, yield, and WUE of the AquaCrop and DSSAT models and the observed values are presented in Figure 5. The AquaCrop model underestimated the biomass by 0.41% and 1.97% in the NM-0.9 *Ep* and M-0.5 *Ep* treatments, and underestimated the yield by 1.30% and 2.35% in the same treatments, respectively. In the M-0.9 *Ep* treatment, it overestimated the biomass by 0.63% and the yield by 1.16%. On the other hand, the DSSAT model underestimated the biomass by 2.09%, 2.54%, and 4.68%, and underestimated the yield by 2.30%, 2.76%, and 7.59% in the aforementioned treatments, respectively.

The relative errors between the simulated values and observed values for biomass, yield, and WUE by both models are shown in Table 6.

The relative errors (RE) between the simulated values and observed values for biomass (B), yield (Y), and *WUE* by the AquaCrop and DSSAT models under three different treatments were as follows: AquaCrop model: RE for B: 0.4%, 0.6%, 2%; RE for Y: 1.3%, 1.2%, 2.3%; RE for WUE: 9.0%, 1.5%, 3.9%. DSSAT model: RE for B: 2.1%, 2.5%, 4.7%; RE for Y: 2.3%, 2.7%, 7.6%; RE for *WUE*: 10.4%, 2.9%, 6.2%. The AquaCrop model exhibited lower RE values for biomass and yield compared to the DSSAT model (Table 6), indicating higher simulation accuracy and better applicability of the AquaCrop model for biomass and yield under the three different cultivation conditions. Among the three treatments, it was observed that the M-0.5 *Ep* environment had the largest errors in biomass and yield simulation compared to the observed values, while the other treatments had relatively smaller errors. Additionally, the simulated values of water use efficiency (WUE) were close to the observed values, with the smallest difference observed in the M-0.9 *Ep* treatment, only 0.03 kg/m^3^, with an RE of 1.5%. The RE for WUE in the WM-0.9 *Ep* and M-0.5 *Ep* treatments were 9.0% and 3.9%, respectively.

### 4.4. Scenario Prediction

Based on the simulation results of various indicators under different treatments, the AquaCrop model showed slightly better simulation performance than the DSSAT model. Therefore, by setting up multiple water scenarios, we can analyze the yield and WUE of the AquaCrop model under different scenarios, providing a basis for exploring the optimal irrigation amount in greenhouses. The scenarios were set as follows: 0.5 *Epan* with and without mulching, 0.6 *Epan* with and without mulching, 0.7 *Epan* with and without mulching, 0.8 *Epan* with and without mulching, 0.9 *Epan* with and without mulching, 1.0 *Epan* with and without mulching, 1.1 *Epan* with and without mulching, 1.2 *Epan* with and without mulching. The simulated yield (Y) and WUE under these scenarios are shown in Figure 6.

It can be observed that the tomato yield increases with the increase in irrigation amount, but excessive irrigation can lead to a decrease in yield. Many studies have confirmed that excessive irrigation can harm crop root systems or microbial activity, leading to reduced productivity [30,31,32]. Under different irrigation levels, the mulching treatment showed higher yield and WUE. However, S15 and S16 treatments were different from other treatments, in that the mulching treatment had a lower yield compared to the non-mulching treatment. This indicates that excessive irrigation has a stronger inhibitory effect on yield than the promoting effect of mulching.

Based on the yield and WUE predictions heatmap distribution under different scenarios (Figure 6), it can be observed that the S7 treatment (0.8 *Epan* with mulching) showed the best simulation performance for greenhouse tomato yield and water use efficiency in the North China region, with values of 8.201 t/ha and 2.79 kg/m^3^, respectively. This suggests the most suitable greenhouse tomato management practice for the North China region, providing valuable insights for tomato cultivation management in the area.

## 5. Discussion

### 5.1. Simulation of Canopy Cover by Using AquaCrop and DSSAT Models

Canopy coverage is an important indicator in crop models. Both models show good simulation performance for canopy coverage. As the growing stage progresses, the simulated values of canopy coverage from both models have small deviations from the observed values in the early and late stages of growth, within 4% deviation. However, during the rapid growth period (11–40 days after transplantation), some differences occur, with a maximum deviation of about 10.2%. There are two possible reasons for this deviation. Firstly, it could be because the model allows input values ranging from 0 to 50 cm^2^/plant for related parameters, but the actual canopy size of individual plants at the time of transplantation is larger. Secondly, the model takes into account the physiological decline of leaves starting from the seedling stage.

### 5.2. Simulation of Soil Moisture, Yield, and Biomass by Using AquaCrop and DSSAT Models

When using the AquaCrop model to simulate soil moisture, yield, and biomass under different water and mulching treatments, it was found that the simulation accuracy of the low water treatment (M-0.5 *Ep* treatment) was slightly lower compared to the high water treatment [33]. This could be attributed to the fact that AquaCrop, as a representative model driven by water, may make incorrect judgments on crop stress responses under severe water stress [34]. This situation became more pronounced during crop aging, as observed in the studies by Heng L [35] and Todorovic M [36]. Stress affects canopy growth and transpiration, and the presence of stress factors during model operation can impact the simulation accuracy of canopy growth, crop transpiration, biomass, and yield, leading to larger simulation errors. In this study, the simulation accuracy of the low water mulching treatment using the DSSAT model was relatively low, consistent with the findings of Yue Yang [37], who used the DSSAT model to simulate the growth and development of dryland wheat and maize and found good overall simulation accuracy. However, in the mulching treatment, the simulation accuracy of the low water treatment was poorer, compared to the high water treatment. Although the DSSAT model underwent corresponding parameter calibration for different treatments, in the high water treatment, based on the simulation results of canopy coverage, soil water content, and biomass accumulation during the tomato growth period, the simulation accuracy of the NM-0.9 *Ep* treatment was slightly higher than that of the M-0.9 *Ep* treatment, indicating that the DSSAT model yields more stable simulation results without mulching. Considering the influence of mulching on maize growth and development, Gao Y [38] improved the DSSAT model by introducing improvements to the evapotranspiration module based on mulching ratio and enhancing the soil temperature module with a soil temperature compensation coefficient based on the crop growing degree-day theory. The improved DSSAT model effectively enhanced the simulation accuracy of summer maize growth, development, and soil water content under mulching conditions. 

### 5.3. Limitations and Suggestions

Although there is an acceptable fit between the simulated results of AquaCrop and DSSAT models and the observed values in simulating the canopy coverage, soil moisture, biomass, and yield of greenhouse tomatoes under different moisture and mulching conditions, there are still certain estimation errors between the simulated and observed values. The AquaCrop model performs significantly when water stress occurs, while DSSAT model performs significantly under mulching conditions. The study found that both the AquaCrop and DSSAT models have lower simulation accuracy under low water (0.5 *Ep*) conditions, indicating that appropriately increasing the irrigation amount of greenhouse tomatoes can reduce simulation errors of the models and improve simulation accuracy. In future research, improvements can be made to the DSSAT model to further enhance the simulation accuracy of tomato growth and development under mulching conditions.

## 6. Conclusions

Under different water and film covering treatments, both models exhibited high simulation accuracy for the canopy coverage index (*k_CC_*) of greenhouse tomatoes. The AquaCrop model had a root mean square error (RMSE) of less than 6.8% between the simulated and observed values of k_CC_, while the DSSAT model had an RMSE of less than 8.5% for the simulated and observed values of k_CC_.

The AquaCrop model has an RMSE of less than 17.96 mm and a normalized root mean square error (NRMSE) of less than 30.51% for soil moisture content at depths of 20 cm, 40 cm, and 60 cm under different moisture and film covering treatments. It also exhibited a consistency index (d_IA_) greater than 0.88 and an efficiency index (EF) greater than 0.52. The DSSAT model had an RMSE of less than 26.95 mm and an NRMSE of less than 28.19% for the same soil water content measurements. It had a *d_IA_* greater than 0.71 and an EF greater than 0.49. Additionally, the AquaCrop model showed higher simulation accuracy under high-water treatments, while the DSSAT model performed better when there was no film covering the soil.

The AquaCrop model exhibited relative errors (RE) of less than 2.0%, 2.3%, and 9.0% for simulated and observed values of biomass (B), yield (Y), and water use efficiency (WUE) under different treatments. In comparison, the DSSAT model had relative errors (RE) of less than 4.7%, 7.6%, and 10.4% for the same variables (B, Y, and WUE).

Based on the simulation results of various indicators under different treatments, the AquaCrop model showed slightly better performance in simulating the growth, development, and soil moisture of greenhouse tomatoes in the North China region compared to the DSSAT model. Using the AquaCrop model, scenario predictions were conducted for 16 different moisture and film covering treatments (S1–S16). From the simulation results, it can be concluded that the S7 treatment (0.8 *Epan* with film covering) had the highest yield and water use efficiency (WUE) for greenhouse tomatoes in the North China region, with values of 8.201 t/ha and 2.79 kg/m^3^, respectively.

## Figures and Tables

**Figure 1 plants-12-03863-f001:**
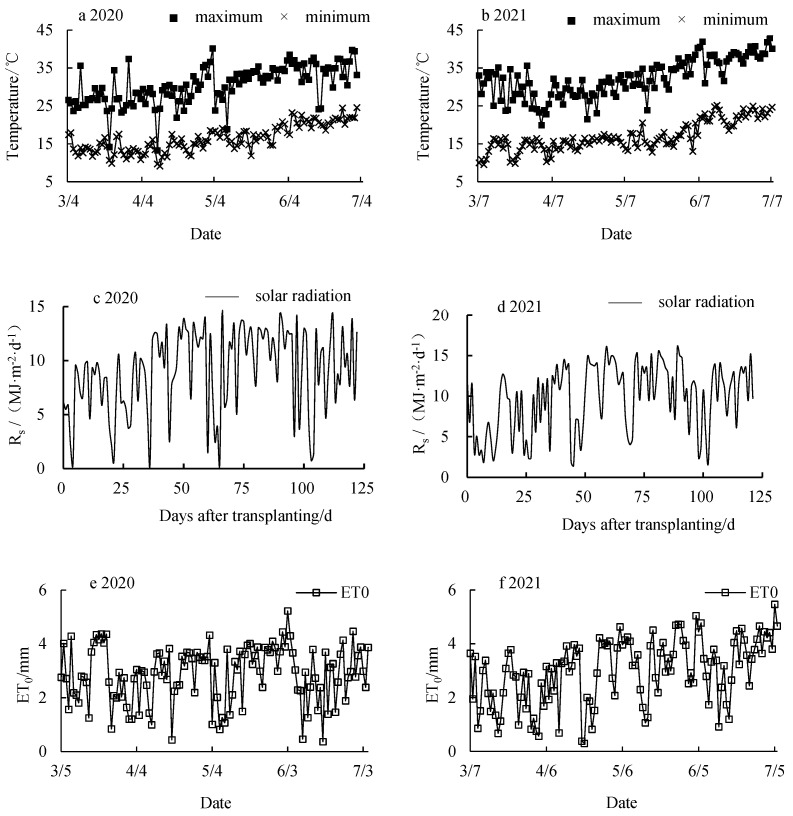
Variation of maximum air temperature, minimum air temperature, and ET_0_ inside the greenhouse in 2020 (**a**,**c**,**e**) and 2021 (**b**,**d**,**f**).

**Figure 2 plants-12-03863-f002:**
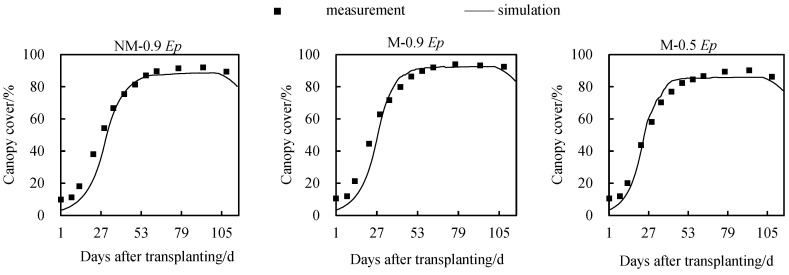
Simulation of canopy coverage under different cultivation conditions by calibrated AquaCrop model.

**Figure 3 plants-12-03863-f003:**
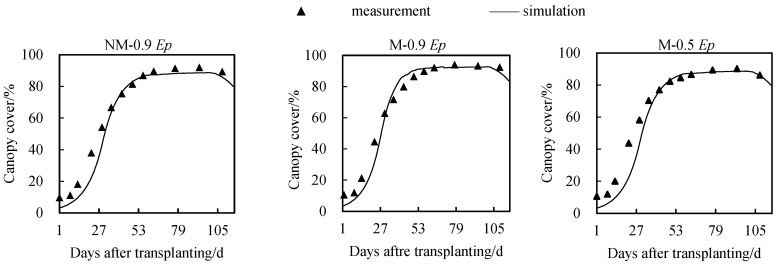
Simulation of canopy coverage under different cultivation conditions by calibrated DSSAT model.

**Figure 4 plants-12-03863-f004:**
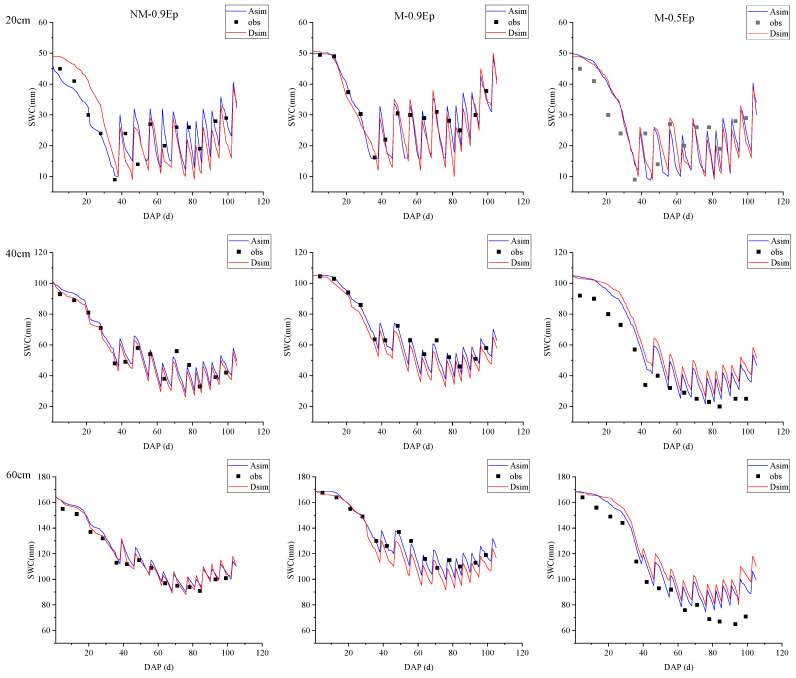
Soil water content under different cultivation conditions simulated by calibrated AquaCrop and DSSAT models.

**Figure 5 plants-12-03863-f005:**
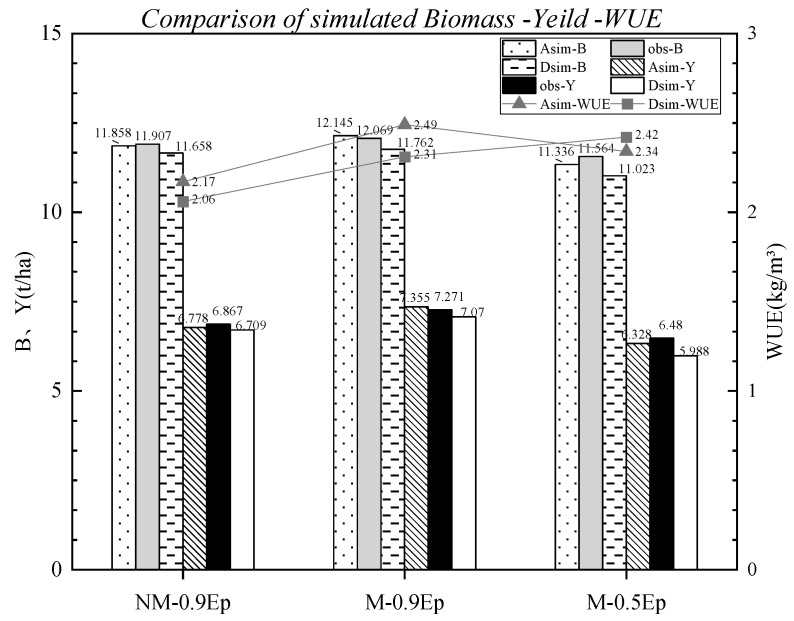
Comparison between the simulated and observed values of biomass, yield, and WUE of calibrated AquaCrop and DSSAT models under three different cultivation conditions.

**Figure 6 plants-12-03863-f006:**
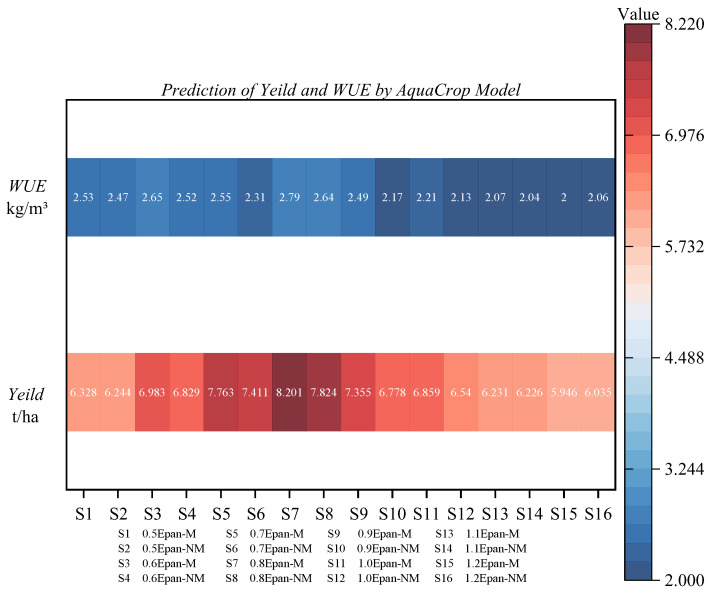
Heat map distribution of greenhouse tomato yield and WUE predicted value under different scenarios of AquaCrop model.

**Table 1 plants-12-03863-t001:** Physicochemical properties and characteristic parameters of the soil.

Soil Depth/cm		0–20	20–40	40–60	60–80	80–100
Particle composition/%	clay particles	21.6	21.4	20.8	11.2	10.4
	silt particles	78.4	78.6	78	78.1	69.3
	sand particles	0	0	1.2	10.7	20.3
soil bulk density/(g·cm^−3^)		1.47	1.44	1.52	1.54	1.46
field capacity/(cm^3^·cm^−3^)		0.306	0.2749	0.3191	0.3756	0.3288
permanent wilting point/(cm^3^·cm^−3^)		0.0806	0.0952	0.0931	0.0723	0.0931
soil organic carbon/%		1	0.5	0.5	0.4	0.3
pH		8.47	8.8	8.82	8.84	8.85
saturation water content/(cm^3^·cm^−3^)		0.4105	0.4209	0.2578	0.4260	0.4292
saturated hydraulic conductivity/(cm·d^−1^)		45.60	25.13	55.56	7.49	130.3

**Table 2 plants-12-03863-t002:** Calibrated parameters of the AquaCrop model.

Parameters	Unit	Type of Parameters	Type of Parameters
Type of crop		Fruit crops	
T_base_	°C	10	conservative parameters
T_upper_	°C	30	conservative parameters
Planting method	transplantation		
Maximum Canopy Coverage CC_X_	%	96	
Maximum Effective Root Depth Z_x_	m	0.9	
Minimum Effective Root Depth Z_n_	m	0.15	
Water Productivity standardized based on ET_0_ and CO_2_	g·m^−2^	15.0	conservative parameters
Reference Harvest Index HI_0_	%	60	
Period of the Harvest Index	d	73	
Canopy Expansion	Moderate tolerance to water stress		
P_exp,upper_		0.15	conservative parameters
P_exp,lower_		0.55	conservative parameters
K_sexp,w_		3.0	conservative parameters
Stomatal Closure	Moderate tolerance to water stress		
Psto		0.50	conservative parameters
Shape factor of stomatal closure water stress coefficient		3.0	conservative parameters
Canopy senescence	Moderate tolerance to water stress		
P_sen_		0.70	conservative parameters
Shape factor of canopy senescence water stress coefficient		3.0	conservative parameters

**Table 3 plants-12-03863-t003:** Calibrated parameters of the DSSAT model.

Definition	Parameters	Unit	Value
time from germination to the appearance of the first flower in terms of light and heat	EM-FL	°C·d^−1^	23.08
time from the first inflorescence flowering to the first inflorescence setting fruit in terms of light and heat	FL-SH	°C·d^−1^	8.41
time from the first inflorescence flowering to the first inflorescence seed production in terms of light and heat	FL-SD	°C·d^−1^	19.37
time from the first inflorescence seed production to physiological maturity in terms of light and heat	SD-PM	°C·d^−1^	47.92
maximum photosynthetic rate of leaves under optimum conditions (measured in terms of CO_2_)	LFMAX	mg·m^−2^·s^−1^	1.50
specific leaf area	SLAVR	cm^2^·g^−1^	316
maximum proportion of dry matter allocated to fruits per day	XFRT		0.62
duration of seed filling in fruits in terms of light and heat	SFDUR	°C·d^−1^	24.15
time required for optimal conditions to achieve the final fruit load in terms of light and heat	PODUR	°C·d^−1^	57.61
ratio of grain mass to fruit mass at fruiting stage	THRESH		8.50

**Table 4 plants-12-03863-t004:** Evaluation index of simulated and measured canopy coverage of AquaCrop and DSSAT models under different cultivation conditions.

Canopy Cover *k_CC_*	NM-0.9 *Ep*		M-0.9 *Ep*		M-0.5 *Ep*	
	AquaCrop	DSSAT	AquaCrop	DSSAT	AquaCrop	DSSAT
RMSE (%)	6.4	6.6	6.8	5.7	5.0	8.5
*d_IA_*	0.98	0.98	0.98	0.98	0.98	0.99

**Table 5 plants-12-03863-t005:** Evaluation indexes of SWC in different soil layers by AquaCrop and DSSAT models under different cultivation conditions.

Soil Depth (cm)	Treatments	AquaCrop				DSSAT			
		RMSE (mm)	NRMSE (%)	*d_IA_*	EF	RMSE (mm)	NRMSE (%)	*d_IA_*	EF
20	NM-0.9 *Ep*	3.18	13.42	0.96	0.83	2.90	9.34	0.97	0.90
	M-0.9 *Ep*	1.18	3.72	0.98	0.98	3.18	11.95	0.96	0.83
	M-0.5 *Ep*	6.23	24.08	0.90	0.52	7.32	28.19	0.86	0.61
40	NM-0.9 *Ep*	4.81	8.44	0.98	0.93	4.26	7.55	0.99	0.95
	M-0.9 *Ep*	2.25	3.24	0.99	0.99	5.56	8.60	0.98	0.92
	M-0.5 *Ep*	13.90	30.51	0.93	0.71	17.65	27.85	0.89	0.49
60	NM-0.9 *Ep*	5.62	4.92	0.98	0.92	3.34	2.84	0.99	0.97
	M-0.9 *Ep*	2.90	2.21	0.99	0.98	7.23	5.74	0.97	0.90
	M-0.5 *Ep*	8.28	13.98	0.97	0.84	19.61	16.22	0.91	0.54

**Table 6 plants-12-03863-t006:** Relative errors between simulated and observed values of biomass, yield, and WUE of AquaCrop and DSSAT models under three different cultivation conditions.

RE (%)	NM-0.9 *Ep*		M-0.9 *Ep*		M-0.5 *Ep*	
	AquaCrop	DSSAT	AquaCrop	DSSAT	AquaCrop	DSSAT
Biomass (B)	0.4	2.1	0.6	2.5	2	4.7
Yield (Y)	1.3	2.3	1.2	2.7	2.3	7.6
*WUE*	9.0	10.4	1.5	2.9	3.9	6.2

## Data Availability

Data are contained within the article.

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
