# Peer review of "Evaluation of Irrigation Modes for Greenhouse Drip Irrigation Tomatoes Based on AquaCrop and DSSAT Models"

_plants, 2023, doi:10.3390/plants12223863_

Round 1

Reviewer 1 Report

Comments and Suggestions for Authors

1. The purpose of this study can be listed in subsections at the end of the introduction, e.g. (1)... ; (2)... ; (3)... , so that it looks a little clearer.

2. It might be better to add a legend to Figure 1.

3. Explain the role of cotton in 2.1 Overview of the experimental area.

4. Please verify the thickness of the soil layer mentioned in the Soil data section of 3.1.2 Parameter Configuration.

5. In 4. Results-4.1 Canopy cover, representative deviations of the simulated and measured values of the two models for canopy cover over the various reproductive periods can be addedFor example, listing the maximum or minimum deviation of the simulated value from the measured value.

6. Why do the two models in 4.2 Soil Water Content only simulate soil moisture in the 0-60 cm layer?

7. In this article, 4.4 Scenario Prediction mentions that Many studies had confirmed that excessive irrigation can harm crop root systems or microbial activity, leading to reduced productivity. List and cite relevant studies here.

8. Please check the words AquaCrop Mode in the comment of Figure 7. mode or model?

9. The Discussion section could discuss canopy cover, soil water content, yield and WUE in parts or paragraphs so that it looks more organized.

10. Please verify the serial numbers of the titles of the Discussion and Conclusions sections.

11. Rich in references, the discussion section can add citations of relevant references to make the article convincing.

12. Please check throughout the article for the presence of upper and lower corner marks and standardize the format of references.

13. Please check the language presentation throughout the article to make the statements more fluent.

Comments on the Quality of English Language

 Minor editing of English language required

Reviewer 2 Report

Comments and Suggestions for Authors

This study evaluated both AquaCrop and DSSAT models in terms of the canopy coverage estimation accuracy. The manuscript is well written, but following minor comments need to be considered in the revised manuscript.

- In Line 320: each validation graph in Figure 4 only include 14 data points, and simulation model shows unnecessary oscillation which may cause an overfitting issue. Is it possible to explain this issue when authors discuss the results of Figure 4?

- In Line 411: Section number should be revised.

- In Line 452: Authors need to compute either MAPE or R2 to address prediction accuracy of the simulation models in terms of %.     
